# Outcomes of Patients with Clinical Stage I-IIIA Large-Cell Neuroendocrine Lung Cancer Treated with Resection

**DOI:** 10.3390/jcm9051370

**Published:** 2020-05-07

**Authors:** Anna Lowczak, Agnieszka Kolasinska-Cwikla, Jarosław B Ćwikła, Karolina Osowiecka, Jakub Palucki, Robert Rzepko, Lidka Glinka, Anna Doboszyńska

**Affiliations:** 1Department of Pulmonology, School of Medicine, University of Warmia and Mazury in Olsztyn, Jagiellonska 78, 11-041 Olsztyn, Poland; anna.doboszynska@wp.pl; 2Department of Oncology and Radiotherapy “Maria Sklodowska-Curie” Memorial Cancer Center, Roentgena 5, 02-781 Warsaw, Poland; adkolasinska@yahoo.com; 3Diagnostic and Therapy Center—Gammed, Lelechowska 5, 02-351 Warsaw, Poland; jbcwikla@interia.pl; 4Department of Cardiology and Internal Medicine, School of Medicine, University of Warmia and Mazury in Olsztyn, Warszawska 30, 10-082 Olsztyn, Poland; 5Department of Psychology and Sociology of Health and Public Health, School of Public Health, University of Warmia and Mazury in Olsztyn, Warszawska 30, 11-041 Olsztyn, Poland; k.osowiecka86@gmail.com; 6Department of Radiology “Maria Sklodowska-Curie” Memorial Cancer Center, Roentgena 5, 02-781 Warsaw, Poland; jmpalucki@gmail.com; 7Pulmonary Hospital in Prabuty, Kuracyjna 30, 82-550 Prabuty, Poland; czita123@poczta.onet.pl; 8Anaesthesiology and Intensive Care Clinical Ward, Clinical University, Hospital in Olsztyn, Department of Anesthesiology and Intensive Care, University of Warmia and Mazury in Olsztyn, Warszawska 30, 10-082 Olsztyn, Poland; lidka.glinka@gmail.com

**Keywords:** Large-cell neuroendocrine carcinoma, survival, resection

## Abstract

Large-cell neuroendocrine carcinoma (LCNEC) is a rare malignancy with poor prognosis. The rationale of the study was to determine the survival of LCNEC patients in I–IIIA clinical stages who underwent resection. A total of 53 LCNEC (89%) and combined LCNEC (11%) patients in stages I–IIIA who underwent surgery with radical intent between 2002–2018 were included in the current study. Overall survival (OS) and time to recurrence (TTR) were estimated. Uni- and multivariable analyses were conducted using Cox-regression model. Patients were treated with surgery alone (51%), surgery with radiochemotherapy (4%), with radiotherapy (2%), with adjuvant chemotherapy (41%), or with neoadjuvant chemotherapy (2%). The median (95% Confidence Interval (CI)) OS and TTR was 52 months (20.1–102.1 months) and 20 months (7.0–75.6 months), respectively. Patients treated in clinical stage I showed better OS than patients in stages II–IIIA (*p* = 0.008). Patients with R0 resection margin (negative margin, no tumor at the margin) and without lymph node metastasis had significantly better TTR. In the multivariate analysis, age was an independent factor influencing OS. Recurrence within 1 year was noted in more than half cases of LCNEC. R0 resection margin and N0 status (no lymph node metastasis) were factors improving TTR. Age >64 years was observed as a main independent factor influencing OS.

## 1. Introduction

Large-cell neuroendocrine carcinoma (LCNEC) is a rare (only 2–3% of all primary lung cancers) and typically aggressive malignancy with poor prognosis [1]. The 5-year overall survival of patients diagnosed with LCNEC ranges from 15% to 57% [2,3,4,5]. The 5-year survival rate in different clinical stages is as follows: for stage I (33–62%), stage II (18–75%), stage III (8–45%), and 0% in patients with stage IV of the disease [2,5,6,7,8]. In the 1970s, neuroendocrine tumors of the lungs were histologically classified into typical carcinoids, atypical carcinoids, and the undifferentiated category represented by small-cell lung cancer (SCLC) [9]. In 1991, Travis et al. [10] introduced a new category of lung cancer: a large-cell neuroendocrine carcinoma (LCNEC) characterized by large cells with a high mitotic rate and neuroendocrine features. The biological and clinical similarities between LCNEC and SCLC have been reported [11,12,13]. LCNEC and SCLC were categorized as high-grade neuroendocrine carcinoma (HGNEC) in the 4th edition of the World Health Organization (WHO) Classification of Lung Tumors [14]. LCNEC often exhibits large-cell morphology and features of neuroendocrine differentiation, including high mitotic rate (>10 mitoses per 10 high-power fields), low nuclear–cytoplasm ratio, and frequent areas of necrosis. The risk factors for LCNEC include age, gender, and exposure to smoking, such that old men (median age 65 years) exposed to heavy smoking are at the greatest risk [15]. Clinicopathological factors that correlate with poor survival rates in published studies include advanced tumor stage, tumor size (greater than 3 cm), and male gender [6,16]. Precise diagnosis of LCNEC is very difficult to achieve preoperatively [17,18]. In most cases, a conclusive pathological diagnosis was obtained from the analysis of resection specimens. That in turn makes it impossible to define a standard of treatment. Primary surgery should be the first option in all operable patients as there is no valid therapeutic approach for LCNEC due to the lack of clinical trials in this setting [19,20]. Most data are based on retrospective analysis. Kujtan L et al. [21] conducted a retrospective evaluation of LCNEC patients with surgically resected stage I of the disease and found that improved survival was achieved by chemotherapy in both stage IA and IB patients.

The aim of this study was to estimate the overall survival (OS) and time to recurrence (TTR) of large-cell neuroendocrine lung cancer (LCNEC) and combined LCNEC patients (combined LCNEC is LCNEC with components of adenocarcinoma, squamous cell carcinoma, or spindle cell carcinoma, and/or giant cell carcinoma [22]) treated with resection in clinical stages I–IIIA. Identification of overall survival (OS) and time to recurrence (TTR) prognostic factors was an additional goal.

## 2. Materials and Methods

A total of 60 cases of patients with a pathological diagnosis of large-cell neuroendocrine lung cancer treated with surgery were recorded in oncological centers located in central and north–eastern region of Poland between January 1, 2002 and December 31, 2018. Among those 60 cases, seven patients in clinical stages IIIB–IV were excluded. The remaining 53 patients with a pathological diagnosis of stage I–IIIA LCNEC and combined LCNEC (large-cell neuroendocrine lung cancer with component of squamous cell carcinoma or adenocarcinoma) were included (Table 1). Patients were treated with radical intention, including surgery alone (27 patients, 51%), surgery and radiochemotherapy (two patients, 4%), surgery and radiotherapy (one patient, 2%), surgery and adjuvant chemotherapy (22 patients, 41%), or surgery and neoadjuvant chemotherapy (one patient, 2%). There were 17 patients that received SCLC-type chemotherapy (platinum/carboplatinum-etoposide; PE/KE schedule) and eight patients were treated with chemotherapy for NSCLC (platinum-vinorelbine; PN schedule). All of the patients were diagnosed based on histopathological confirmation during core needle biopsy, mediastinoscopy, pneumonectomy, lobectomy, segmentectomy, wedge resection, and locoregional lymph nodes resection. Patients included in analysis met the criteria of ≥18 years old, Caucasian race, LCNEC or combined type LCNEC diagnosis, locally advanced clinical stage I–IIIA, and underwent surgery with curative intention. The stage of cancer development was determined using UICC (The Union for International Cancer Control) TNM (T—tumor, N—nodes, M—metastasis) classification of Malignant Tumors—8th edition [23]. The degree of pathomorphic stage (pTNM) was assessed in all patients.

The analysis was retrospective. The data was collected through medical records, hospital databases, and registry office, and was supplemented by interviews with patients, their families, and attending physicians.

The study was approved by the bioethics committee affiliated with the University of Warmia and Mazury in Olsztyn. Approval Number: 12/2018.

## 3. Statistical Analysis

Survival probabilities were estimated by Kaplan–Meier method and the differences in survival (median OS/TTR) were compared using the log-rank test. The percentages of patients with 1-, 3-, and 5-year survival in different clinical subgroups were compared by Fisher’s exact test. Uni- and multivariable predictors of overall mortality were estimated through Cox-regression analysis. Univariate variables with *p* < 0.1 were included in the multivariable model. Overall survival (OS) was defined as the time from histopathological diagnosis till death from any cause or last follow-up censored. Time to recurrence (TTR) was calculated from the date of the beginning of curative treatment to the time of recurrence. Patients, who died without recurrence or who were known to be recurrence-free were censored at the last time of follow-up. A *p* value <0.05 was considered to be significant. The analysis was conducted using TIBCO Software Inc. (2017). Statistica (data analysis software system), version 13. http://statistica.io.

## 4. Results

Overall, a total of 53 patients with LCNEC (*n* = 47) and combined LCNEC (*n* = 6) were included in the analysis. The age range of the patients was 47–79 years with mean age of 62.7 and standard deviation (SD) of 7.3 years. The group of patients consisted of 22 women (42%) and 31 men (58%). Half of the patients were in the IIB–IIIA clinical stage, while the majority of patients were characterized by Ki67 >55 (58%), the lack of lymph node metastasis (64%), and size of tumor ≤4 cm (60%). The size of tumor ranged 7–80 mm. The mean size of tumor was 37.3 (SD 18.6 mm). Twelve patients (23%) underwent non-radical surgery R1-2. Among patients who received chemotherapy, 68% patients were treated with SCLC-type of chemotherapy and 32% patients received NSCLC-chemotherapy (Table 1).

Median follow-up of patients was 29 months (range: 2–169 months). The 1-, 3-, and 5-year OS for all patients was 88%, 59%, and 41%, respectively. The median overall survival was 52 months (95% Confidence Interval (CI): 20.1–102.1 months; Figure 1). 

The median time to recurrence was 20 months (95% CI: 7.0–75.6 months) and 1-, 3-, and 5-year TTR was 58%, 41%, and 31%, respectively (Figure 2).

Patients who underwent surgery with radical intension in clinical stage I showed a significantly better overall survival in comparison with patients treated in stages II–IIIA (*p* = 0.008). We did not discover factors, such as age, lymph node status, or size of tumor to have a significant influence on overall survival of LCNEC patients. The median OS (mOS) was 28 months (95% CI: 19.0–54.4 months) and 64 months (95% CI: 21.2–not reached) for N+ and N–, respectively (*p* = 0.08). In terms of age, mOS for younger patients (≤ 64 years) was 105 months (95% CI: 24.0–not reached) and was better in comparison with older patients (> 64 years) whose mOS was 27 months (95% CI: 16.6–57.7; *p* = 0.06). Patients with tumor ≤4 cm had better prognosis than patients who received treatment on tumor >4 cm (respectively: 54 months mOS (95% CI: 26.3–not reached) and 24 months mOS (95% CI: 15.4–63.6); *p* = 0.09) (Table 2).

The median TTR (mTTR) for R0 vs. R1-2 patients was 31 months (95% CI: 7.5–not reached) vs. 9 months (95% CI: 5.0–22.0), respectively (*p* = 0.05). Patients with N0 status survived longer time torecurrence, such that mTTR was 42 months (7.4–not reached) for N0 patients compared to lymph node metastasis patients with an mTTR of 12 months (95% CI: 6.8–32.3; Table 2). 

The significant difference between percentages of patients with 1- and 5-year survival due to age, clinical stage, lymph node status, radicality of surgical resection, and type of surgery was observed. Younger age (≤64 vs. >64 years), less advanced disease (CS I vs. II-IIIA), absence of lymph node metastasis (N− vs. N+), negative resection margin (R0 vs. R1–2), and less radical surgery (segmental or wedge resection vs. pneumonectomy or lobectomy) significantly increased the percentage of survivors and decreased frequency of recurrence at 5-years (Table 3).

In the univariate analysis, a significant influence on OS was demonstrated for clinical stage (*p* = 0.01). Patients diagnosed with clinical stages II–IIIA were at about 3.5 times higher risk of death than with CS I (HR: 3.4; 95% CI: 1.28–9.14; Table 3). Moreover, in the univariate analysis, a significant influence on TTR was demonstrated for lymph node status and the extent of resection (*p* < 0.05). In both cases the risk of progression increased by twice the amount (Table 4).

In the multivariate analysis, it was determined that type of treatment and age were independent factors influencing OS. The risk of death in older patients increased by more than twice the amount (≤64 years vs. >65 years, HR: 2.36; 95% CI: 1.02–5.48; *p* = 0.04). Almost three times more risk of death was observed in patients treated with alone surgery compared to patients that underwent adjuvant or neoadjuvant chemo- and/or radiotherapy (HR: 2.79; 95% CI: 1.18–6.56; *p* = 0.02). No significant factors with an influence on TTR were reported in the multivariate model analysis (Table 5). 

## 5. Discussion

Pulmonary neuroendocrine tumors represent about 20% of all lung cancers [10] and published results often include a small number of patients [17,24]. There are few studies examining the outcomes of surgery in LCNEC patient populations. These studies are rather primarily single- institution analyses limited by small cohort sizes and outdated results [3,5,18,25]. In a recent study Raman et al. [26] performed a retrospective large cohort study including 6092 LCNEC patients with stages I–IIIA using the National Cancer Database to examine the outcomes of surgery by stage. In comparison to the United States, where data about 80% of cancers diagnosed across 1500 centers have been prospectively collected by independent tumor registrars [27], in Poland, the National Cancer Register have unfortunately not collected any data (such as stage of disease) to estimate the survival analyses. To our knowledge there are no published results of LCNEC patients survival outcomes in the Polish population, therefore this subject is worth exploring. In the current study, we conducted a retrospective analysis of 60 patients who underwent a surgical procedure with curative intention. The mean age was above 60 years, which is comparable with other authors [28]. The majority of patients included in our analysis were male (58%), underwent lobectomy (79%), and R0 resection margin was achieved in 77%. The predominance of men was also reported in other studies and R0 resection was observed in 90–99.2% [2,20,28]. Other authors have also confirmed that the most frequently used surgery option was also lobectomy [5,25,26]. All analyzed patients, whose smoking status was known, were smokers. Similar to our results, Roesel et al. [28] also did not report significant association of survival with gender. This is in contrast to another authors, who noted the relation between lower OS and the male gender [20,29]. Although we did not confirm that observation, other authors note significantly better survival in less advanced T-stage and in the absence of lymph node metastasis [20,28,29]. The authors [28] showed the overall 1-, 3-, and 5-year survival rates of 83.7%, 63.2%, and 53.8%, respectively, whereas the results of overall survival obtained in our study were 88%, 59%, and 41% for 1-, 3-, and 5-year survival, respectively. Eichhorn et al. [30] also established that half of the patients treated with surgery for LCNEC survived three years. The median of the overall survival for all patients in stages I–IIIA in our study was 4.3 years and it was longer than the median OS of resected more advanced-stage LCNEC patients estimated by Sarkaria et al. [20], which was 3.4 years. The WHO histopathological diagnosis criteria [14] of LCNEC states that in most cases, recognition is postoperative and that patients with LCNEC generally have poor prognoses of 5-year survival at all stages with survival rates of 15–57% [3,8,29,31,32,33,34]. Even patients with pathological stage I LCNEC have poor prognoses with 5-year survival rates of 27–67% [2,3,5,6,29,32]. Iyoda and colleagues [35] compared the prognoses of LCNEC patients with pathological stage IA to those of patients with adenocarcinomas or squamous cell carcinomas of the same stage and they revealed that the 5-year survival rate of the LCNEC patients was 54.5% in comparison with 89.3% in the other group. In our study we estimated early clinical stage of the disease as a factor with a statistically significant positive influence on OS. The other authors [20,29,34] also determined a significant relationship between survival and clinical stage. We determined that 5-year overall survival of patients who underwent resection was 67% and 26% for clinical stage I vs. II–IIIA, respectively (*p* = 0.008). Similar results (64.5%) were achieved in a single-institution retrospective analysis conducted on 125 patients [36]. In USA Raman et al. [26] performed 5-year survival (50%, 45%, and 36% for CS I, II, and IIIA, respectively) using a largest cohort on the world (>6000 patients). Similar to Eichhorn et al. [30], who determined that advanced nodal status was significantly associated with poor outcome, in our analysis, LCNEC patients treated with resection who did not show lymph node metastasis had better prognosis (OS and TTR) in comparison with N+ patients, although the difference was not statistically significant. 

In our analysis, the percentage of LCNEC patients without recurrence after three and five years was 41% and 31%, respectively. Similar results were obtained by other authors, such that 3- and 5-year recurrence-free survival was reported in 45% and 38–39.3% of patients, respectively [3,30,34]. Many recurrent tumors as distant metastases were observed in LCNEC patients with complete resection [20,21,37]. Therefore, surgery alone could be insufficient to treat patients with LCNEC and adjuvant therapy may be necessary [36,38]. In our analysis half of the patients were treated with surgery alone and 42% of patients were treated with adjuvant chemotherapy. Some authors [6,7,39] reported that postoperative adjuvant chemotherapy was effective in patients with LCNEC. Sarkaria et al. [20] reported a high response rate to neoadjuvant chemotherapy of LCNEC patients and that resected advanced-stage patients receiving a combination of neoadjuvant and/or adjuvant chemotherapy may have a survival advantage. Iyoda et al. [21] noted that 50% of LCNEC patients after surgery had confirmed recurrent tumors upon follow-up examinations and most of them within three years. The authors also noted that patients who underwent platinum-based adjuvant chemotherapy had a significantly lower rate of tumor recurrence and a higher rate of disease-free survival than those who had non–platinum-based adjuvant chemotherapy or no adjuvant chemotherapy. Raman et al. [26] showed that perioperative chemotherapy was associated with improved survival for pathologic stage II–IIIA disease. In our multivariate analysis, next to age >64 years (*p* = 0.04), surgery alone (*p* = 0.02) was found to be associated with an increased risk of death. Generally, neo- or adjuvant therapy could be the most important predictor of better OS, as most older patients were treated with alone surgery (59%) compared to younger patients, who received neo- or adjuvant therapy (69%); this difference was statistically significant (*p* = 0.04). Other authors [3,29] confirm older age as an adverse indicator for survival outcomes of LCNEC patients who underwent surgery. In China, authors provide a multivariate model of predictors of overall survival in pulmonary large-cell neuroendocrine carcinoma and determined that age at diagnosis, gender, tumor stage, N stage, tumor size, and primary site of surgery were independent prognostic factors influencing OS [40]. In a multivariate analysis conducted by Sarkaria and colleagues [20] male gender, advanced-stage at diagnosis, and pulmonary comorbidities remained significant predictors of worse survival.

Owing to the poorly differentiated features, LCNEC and SCLC were classified as the high-grade NETs of the lung. LCNEC was a rare histologic type of lung cancer with an incidence of approximately 3%, while SCLC was a common histologic type of lung cancer accounting for 15–20% of all lung cancers [1,41]. Compared with carcinoid, LCNEC and SCLC had higher mitotic rates, more necrosis and poorer prognosis, and could even manifest combined with other lung cancer types [42]. Although LCNEC and SCLC shared several similar clinical and histologic features, questions remained as to whether it was reasonable to classify LCNEC and SCLC within the same category. Wang et al. [43] compared the clinicopathological characteristics and survival outcomes between LCNEC and SCLC in the large number of patients (*n* = 19,405) to improve understanding of high-grade NETs of the lung. The authors determined that high-grade LCNEC patients had a better overall survival and cancer-specific survival than high-grade SCLC patients also in the regional stage, distant stage, and surgery subgroups. Raman et al. [26] also confirmed better survival prognosis by stage I–IIIA of patients with LCNEC compared to SCLC. Chemotherapy treatment for LCNEC is a subject of debate since it seems to be less chemosensitive than SCLC. In the American Society of Clinical Oncology (ASCO) guideline, either platinum–etoposide chemotherapy (SCLC-PE) treatment or the same regimen as for non-small-cell non-squamous-cell carcinoma is advised for LCNEC [44], although SCLC-PE is considered as the most appropriate [44]. Nevertheless, recent studies indicate that patients with LCNEC have a more favorable outcome when treated with platinum–gemcitabine or taxane chemotherapy (NSCLC-GEM/TAX) compared with SCLC-PE [45,46,47]. We compared survival of patients treated with surgery and chemotherapy regimens derived from SCLC protocols (platinum+etoposide) and NSCLC protocols (platinum+vinorelbine) and we did not detect a significant difference in treatment results and patient outcome. It could be caused by small size of analyzed subgroups. Some small population-based studies showed that advanced LCNEC patients benefited from SCLC-based chemotherapy, rather than SCLC-based chemotherapy plus non-small-cell lung cancer (NSCLC)-based chemotherapy, and showed similar survival outcomes to those found in advanced SCLC [48,49,50,51]. The molecular characteristics that may explain these differences in the response to different chemotherapies remain unknown. Derks et al. [52] tried to assess the predictive role of *RB1* mutation on chemotherapy outcomes. The authors concluded that LCNEC patients with wild-type *RB1* gene or express the RB1 protein had a significantly longer OS with NSCLC-GEM/TAX treatment than with SCLC-PE chemotherapy. 

Our study is a retrospective cohort analysis limited by small number of patients and heterogeneity of clinical stage and methods of treatment. We analyzed groups of patients treated with surgery with intention to treat among 132 LCNEC/combined LCNEC patients treated with radical, palliative, or symptomatic intention. There are lack of results showing LCNEC survival outcomes of the Polish population and this area of research should be expanded in the future. 

## 6. Conclusions

More than half of LCNEC patients who underwent surgery showed recurrences within one year. Patients with R0 resection margin and without lymph node metastasis had significantly better TTR. Although advanced clinical stage (I vs. II–IIIA) was noted as an adverse prognostic factor, the age ≤64 years and neo- or adjuvant therapy were observed as beneficial independent factors influencing OS. 

## Figures and Tables

**Figure 1 jcm-09-01370-f001:**
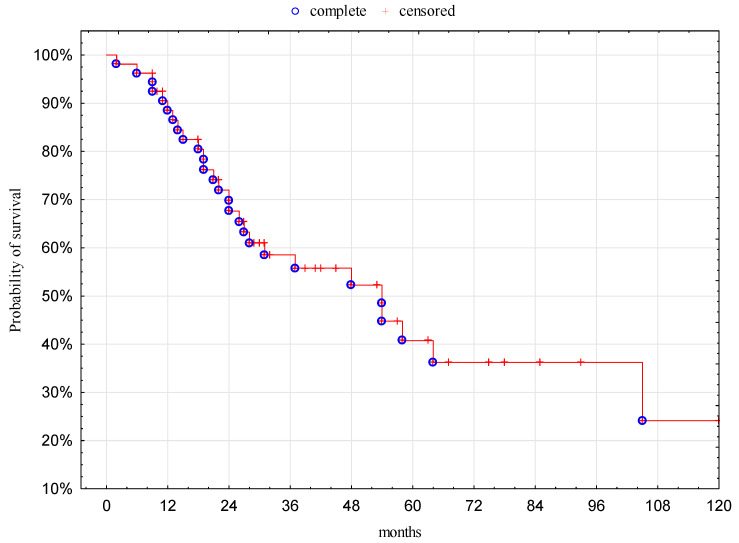
Overall survival of patients.

**Figure 2 jcm-09-01370-f002:**
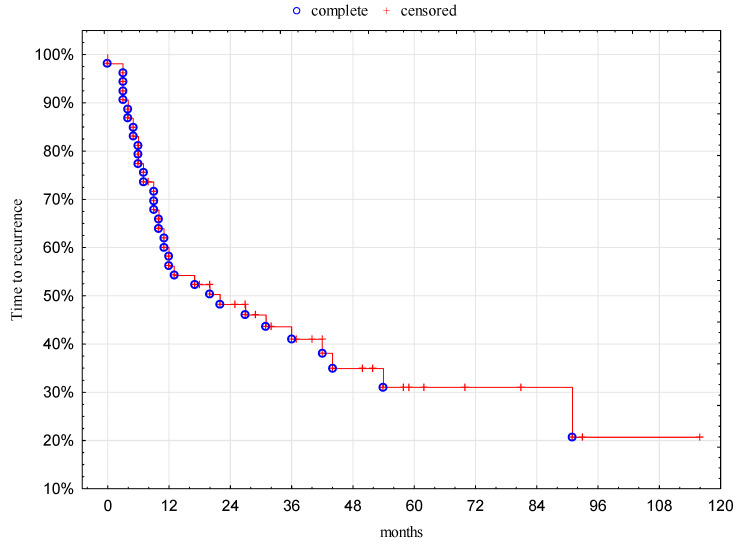
Time to recurrence.

**Table 1 jcm-09-01370-t001:** Patient characteristics.

		*N*	%
**All**		53	100
**Age [years] (range: 47–79, mean 62.7 ± 7.3)**			
	≤64	29	55
	>64	24	45
**Gender**			
	women	22	42
	men	31	58
**Smoking status [pack-years] (range: 12–100, mean 42.4 ± 19.4)**			
	smoker	43	81
	non-smoker	0	0
	no data	10	19
**Clinical stage**			
	IA	16	30
	IB	5	9.5
	IIA	5	9.5
	IIB	12	23
	IIIA	15	28
**Lymph nodes**			
	N0	34	64
	N1	11	21
	N2	8	15
**Ki67**			
	≤55	18	34
	>55	31	58
	no data	4	8
**Localization**			
	left lung	25	47
	right lung	28	53
**Radicality of surgical resection**			
	R0	41	77
	R1–2	12	23
**Size of tumor [mm] (range: 7–80, mean 37.3 ± 18.6)**			
	≤4 cm	32	60
	>4 cm	21	40
**Size of tumor**			
	≤3 cm	24	45
	>3–5 cm	19	36
	>5–7 cm	8	15
	>7 cm	2	4
**Type of surgery**			
	pneumonectomy	3	5.5
	bilobectomy	1	2
	lobectomy	42	79
	segmentectomy	5	9.5
	wedge resection	2	4
**Type of treatment**			
	alone surgery	27	51
	surgery with adjuvant chemotherapy	22	41
	surgery with neoadjuvant chemotherapy	1	2
	surgery with radiochemotherapy	2	4
	surgery with radiotherapy	1	2
**Type of chemotherapy**			
	SCLC-chemotherapy	17	68
	NSCLC-chemotherapy	8	32


± standard deviation. N0—no lymph node metastasis; N1—metastasis in ipsilateral peribronchial and/or hilar lymph nodes and intrapulmonary nodes; N2—metastasis in ipsilateral mediastinal and/or subcarinal lymph node(s). Ki67—a proliferation marker to measure the growth fraction of cells in human tumors. R0—negative margin, no tumor at the margin; R1—microscopic positive margin, tumor identified microscopically at the margin; R2—macroscopic positive margin, tumor identified grossly at the margin. SCLC (small-cell lung cancer)—type chemotherapy (platinum/carboplatinum-etoposide; PE/KE schedule); NSCLC (non-small-cell lung cancer)—type chemotherapy (platinum-vinorelbine; PN schedule).

**Table 2 jcm-09-01370-t002:** Differences in overall survival and time to recurrence of long-cell neuroendocrine carcinoma (LCNEC) cancer patients due to prognostic factors.

Variables		Overall Survival (OS)			Time to Recurrence (TTR)		
		Median OS (95% CI Months)		Log-Rank Test	Median TTR (95% CI Months)		Log-Rank Test
**All**		52	(20.1–102.1)	*p*	20	(7.0–75.6)	*p*
**Age**							
	≤64	105	(24.0–not reached)	0.06	31	(9.0–not reached)	0.16
	>64	27	(16.6–57.7)		12	(5.0–43.5)	
**Gender**							
	women	48	(21.1–83.4)	0.78	22	(7.4–45.9)	0.57
	men	54	(18.5–not reached)		13	(6.8–not reached)	
**Clinical stage**							
	I	105	(33.4–not reached)	0.008	91	(6.0–not reached)	0.08
	II–IIIA	28	(17.5–58.6)		13	(7.0–42.0)	
**Tumor status**							
	T1-2	54	(22.9–98.8)	0.81	27	(7.0–66.1)	0.57
	T3	21	(13.0–not reached)		12	(7.0–not reached)	
**Lymph nodes status**							
	N+	28	(19.0–54.4)	0.08	12	(6.8–32.3)	0.05
	N–	64	(21.2–not reached)		42	(7.4–not reached)	
**Ki67**							
	≤55	58	(20.0–71.5)	0.83	20	(5.5–57.4)	0.89
	>55	48	(19.5–not reached)		22	(9.0–not reached)	
**Localization**							
	left lung	58	(22.0–not reached)	0.54	27	(7.0–not reached)	0.99
	right lung	48	(17.4–90.1)		20	(5.0–74.6)	
**Radicality of surgical resection**							
	R0	58	(20.9–not reached)	0.28	31	(7.5–not reached)	0.05
	R1–2	27	(16.5–46.9)		9	(5.0–22.0)	
**Size of tumor**							
	≤4 cm	54	(26.3–not reached)	0.09	31	(6.0–79.3)	0.58
	>4 cm	24	(15.4–63.6)		13	(7.5–not reached)	
**Type of surgery**							
	pneumonectomy and lobectomy	54	(19.2–92.5)	0.92	27	(8.0–86.4)	0.15
	segmental and wedge resection	26	(20.5–58.0)		9	(3.8–20.4)	
**Type of treatment**							
	alone surgery	37.1	(13.1–54.0)	0.09	13	(4.8–49.7)	0.14
	surgery and neo- or adjuvant therapy	63.9	(24.0–not reached)		36	(9.4–not reached)	
**Type of chemotherapy**							
	SCLC-chemotherapy	64	(24.0–not reached)	0.82	42	(7.9–not reached)	0.84
	NSCLC-chemotherapy	not reached	(15.0– not reached)		12	(9.0–not reached)	

T1—tumor ≤3 cm; T2—tumor >3 to ≤5 cm; T3—tumor >5 to ≤7 cm. N (-)—no lymph node metastasis; N (+)—metastatic lymph nodes. SCLC (small-cell lung cancer)—type chemotherapy (platinum/carboplatinum-etoposide; PE/KE schedule); NSCLC (non-small-cell lung cancer)—type chemotherapy (platinum-vinorelbine; PN schedule).

**Table 3 jcm-09-01370-t003:** The percentage distribution of patients with 1-, 3-, and 5-year survival in different clinical subgroups.

Variables		Overall Survival (OS)				Time to Recurrence (TTR)			
		1-year (%)	3-year (%)	5-year (%)	Fisher’s Exact Test	1-year (%)	3-year (%)	5-year (%)	Fisher’s Exact Test
**All**		88	59	41	*p* *	58	41	31	*p* *
**Age**									
	≤64	93	67	51	0.04	65	42	42	0.01
	>64	83	48	24		46	42	11	
**Gender**									
	women	86	60	41	0.89	63	33	27	0.16
	men	90	58	41		52	48	35	
**Clinical stage**									
	I	90	78	67	0.001	61	61	51	0.006
	II–IIIA	87	47	26		58	33	20	
**Tumor status**									
	T1–2	87	62	41	0.52	60	42	28	0.08
	T3	87	50	50		62	50	50	
**Lymph nodes status**									
	N+	89	44	13	<0.001	47	21	14	0.02
	N–	89	68	57		61	55	44	
**Ki67**									
	≤55	83	59	35	0.68	61	44	35	1.00
	>55	90	55	43		54	42	31	
**Localization**									
	left lung	96	62	43	0.89	56	41	26	0.43
	right lung	82	56	39		56	40	35	
**Radicality of surgical resection**									
	R0	90	64	46	0.01	60	49	41	<0.001
	R1-2	81	38	19		50	24	8	
**Size of tumor**									
	≤4 cm	90	69	43	0.79	59	44	32	0.87
	>4 cm	86	43	37		52	37	30	
**Type of surgery**									
	pneumonectomy and lobectomy	87	60	44	0.02	58	43	35	<0.001
	segmental and wedge resection	100	50	25		43	29	not reached	
**Type of treatment**									
	alone surgery	77	54	22	0.05	52	35	23	0.27
	surgery and neo- or adjuvant therapy	100	64	52		60	48	39	
**Type of chemotherapy**									
	SCLC-chemotherapy	100	63	55	0.56	64	51	43	0.77
	NSCLC-chemotherapy	100	63	63		50	38	38	

* *p*-value—comparison the percentages between 1-year OS/TTR and 5-year OS/TTR-.

**Table 4 jcm-09-01370-t004:** Univariate Cox regression.

Variables		Overall Survival (OS)			Time to Recurrence (TTR)		
		HR (95% CI)		*p*	HR (95% CI)		*p*
**Age**	≤64	1.00	Reference		1.00	Reference	
	>64	2.12	(0.97–4.62)	0.06	1.63	(0.82–3.24)	0.16
**Gender**	women	1.00	Reference		1.00	Reference	
	men	0.90	(0.42–1.93)	0.79	0.83	(0.42–1.62)	0.58
**Clinical stage**	I	1.00	Reference		1.00	Reference	
	II-IIIA	3.43	(1.28–9.14)	0.01	0.51	(0.24–1.11)	0.09
**Tumor status**	T1–2	1.00	Reference		1.00	Reference	
	T3	1.15	(0.39–3.38)	0.80	0.75	(0.26–2.15)	0.59
**Lymph nodes status**	N−	1.00	Reference		1.00	Reference	
	N+	2.00	(0.92–4.36)	0.08	1.96	(0.99–3.91)	0.05
**Ki67**	≤55	1.00	Reference		1.00	Reference	
	>55	0.92	(0.41–2.06)	0.84	0.95	(0.46–1.97)	0.89
**Localization**	left lung	1.00	Reference		1.00	Reference	
	right lung	1.26	(0.59–2.71)	0.55	1.01	(0.51–1.98)	0.99
**Radicality of surgical resection**	R0	1.00	Reference		1.00	Reference	
	R1–2	1.67	(0.70–3.99)	0.25	2.08	(1.01–4.31)	0.047
**Size of tumor**	≤4 cm	1.00	Reference		1.00	Reference	
	>4 cm	1.97	(0.91–4.27)	0.08	1.22	(0.61–2.43)	0.58
**Type of surgery**	pneumonectomy and lobectomy	1.00	Reference		1.00	Reference	
	segmental and wedge resection	1.06	(0.36–3.08)	0.92	2.04	(0.84–4.98)	0.12
**Type of treatment**	surgery and neo- or adjuvant therapy	1.00	Reference		1.00	Reference	
	alone surgery	1.93	(0.89–4.19)	0.09	1.68	(0.84–3.33)	0.14
**Type of chemotherapy**	NSCLC-chemotherapy	1.00	Reference		1.00	Reference	
	SCLC-chemotherapy	1.17	(0.31–4.41)	0.82	0.90	(0.30–2.69)	0.85

**Table 5 jcm-09-01370-t005:** Multivariate Cox regression.

Overall Survival (OS)					Time to Recurrence (TTR)				
Variables		HR (95% CI)		*p*	Variables		HR (95% CI)		*p*
**Age**	≤64	1.00	Reference						
	>64	2.36	(1.02–5.48)	0.04					
**Type of treatment**	surgery and neo- or adjuvant therapy	1.00	Reference						
	alone surgery	2.79	(1.18–6.56)	0.02					
**Clinical stage**	I	1.00	Reference		**Clinical stage**	I	1.00	Reference	
	II–IIIA	3.42	(0.67–17.47)	0.14		II–IIIA	1.59	(0.60–4.17)	0.35
**Lymph nodes status**	N–	1.00	Reference		**Lymph nodes status**	N−	1.00	Reference	
	N+	1.81	(0.56–5.86)	0.32		N+	1.19	(0.45–3.18)	0.72
**Size of tumor**	≤4 cm	1.00	Reference		**Radicality of surgical resection**	R0	1.00	Reference	
	>4 cm	0.98	(0.31–3.14)	0.97		R1-2	1.71	(0.73–4.03)	0.22

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
