# Peer review of "Outcomes of Patients with Clinical Stage I-IIIA Large-Cell Neuroendocrine Lung Cancer Treated with Resection"

_jcm, 2020, doi:10.3390/jcm9051370_

Round 1

Reviewer 1 Report

Dear Authors,

This paper is a well written overview of patients with LCNEC treated with surgery in Poland. Differences could be found between stage I and II-IIIA based on survival. The influence of adjuvant chemotherapy cannot be drawn.

However I have some major concerns: This is a small cohort and the paper from Raman (JTO 2019)  is not taken into account in the discussion. This is a larger series (>6000) which describes the same item.
Maybe add therefore that this is a Polish cohort.

Besides Derks (CCR 2018) has described influence of different staining's to cover optimal chemotherapy strategy. This has not been discussed at all.

Further comments are:

1. In the introduction (line 66) the definition of combined LCNEC is not given, This should be mentioned

2. In the materials (line 78) What is a tumorectomy. Should this be lobectomy or pneumonectomy. Maybe use standard words in stead and then leave out surgery.

3. It seems as this is a retrospective analysis. However this is not clearly stated.

4. In Table 1 clinical stage is not adding up to 100%
5. What do you mean with treatment with chemotherapy SCLC/NSCLC? There were only 6 combined LNCEC, so 17 SCLC Is strange.

6. In the results it is not clear what the +7.3years is; confidence interval, IQ range...? (line 98) same for tumor size (line 101)

7. I miss a legend in Figure 1 and 2; what are the red crosses and what are the blue circles?

8. In lines 113-121  I miss confidence intervals for survival

Author Response

Thank you for a review and valuable comments.

We cited in discussion part the paper from Raman (JTO 2019), which summarize the largest on the world group of LCNEC patients in line: 188-192 and 224-226 and 246-252 and 270-272.

We cited the paper, which described the predictive role of RB1 mutation on chemotherapy outcomes in discussion part in line: 286-289.

Ad.1 In the Introduction we defined combined LCNEC in line: 69-70.

Ad.2 In the materials and methods part we have highlighted type of surgeries using standard words instead of “surgery” in line: 85-86 and we added the percentage of different types of surgery in table 1 with characteristic group.

Ad.3 We stated that our analysis is retrospective in materials and methods part in line 93.

Ad.4 In table 1 we added up clinical stage variable to 100%.

Ad.5 We explained the meaning of type of chemotherapy: SCLC-chemotherapy and NSCLC-chemotherapy in materials and methods part in line: 81-83 and we did some changes in tables.

Ad.6 The ± meant standard deviation. We explained that sign under the table 1 and did some changes in results in line: 115, 119.

Ad.7 We corrected the figures 1 and 2. The red crosses were censored data and the blue circles were completed data.

Ad. 8 We completed missed confidence intervals to median survival in line: 139-144, 150-154.

Reviewer 2 Report

Does not provide any new information from what is known. 60 patients is a small number.

Author Response

Thank you for a review and valuable comment.

There are few studies examining the outcomes of surgery in LCNEC patients population. This studies are rather primarily single- institution analyses limited by small cohort sizes and old results. In Poland National Cancer Register unfortunately have not collected any data like stage of disease to estimate the survival analyses. To our knowledge there is no published results of LCNEC patients survival outcomes in polish population therefore this subject is worth exploring.

Our study is a retrospective cohort analysis limited by small number of patients. Although we selected to analysis group of patients treated with surgery with intention to treat among 132 LCNEC/combined LCNEC patients treated with radical, palliative or symptomatic intention.

We included in discussion part the information about limitation our study in line: 300-304.

Reviewer 3 Report

Original study by Lowczak A et al. entitled "Outcomes of patients with clinical stage I-IIIA large cell neuroendocrine lung cancer treated with resection" is an interesting, but not very original, observation. In univariate analysis, the authors proved that the time to recurrence (TTR) depends on the radicality of the surgical resection and the presence of metastases in the lymph nodes, while the overall survival depends on clinical stage of disease. The results are not surprising, but important, given the rarity of neuroendocrine large cell lung cancer. However, the article requires significant additions.

Major revision

  1. The study group is small and heterogeneous in terms of disease stage and treatment methods. Authors should point out this weakness of the study in the discussion.
  2. The analysis of overall survival and time to recurrence lacked such an important factor as the type of pre- and postoperative treatment. The possibility of using neoadjuvant or adjuvant chemotherapy or radiotherapy is an extremely important factor determining the survival of patients with surgically resected lung cancer. Another omitted prognostic factor is a type of surgical resection (lobectomy, pneumonectomy, wedge resection)
  3. It is surprising that in multivariate analysis only the age of the patients showed an impact on the overall survival, while in univariate analysis it was the disease stage. This may suggest that older patients were treated differently than younger patients (e.g. they were less likely to receive adjuvant treatment, or the extent of surgical resection was smaller).

Minor revision

  1. The authors use the term “progression free survival (PFS)”. This term is rather reserved for systemic treatment, where partial remission of the disease could be observed, followed by progression.In surgically resected patients, the term “time to recurrence” should be used.
  2. The term "average" used by the authors is imprecise. In fact, the authors use the mean and standard deviation.
  3. In Table 2, it is not known what the statistical significance analysis relates to: median overall survival or 1-, 3- and 5-year survival. It should be guessed that this applies to mOS.In that case HR and 95% CI calculated by Kaplan-Meier method should appear here.The percentages of patients with 1-, 3- and 5-year survival from different clinical groups could be compared by Fisher exact test.

Author Response

Thank you for a review and valuable comments.

Major revision:

Ad.1 We provided a retrospective analysis, multicentral. We agree that the size of analyzed group and heterogeneity in terms of clinical stage and treatment method are the limitation of our study. We stated that in discussion part in line: 300-304.

Ad.2, 3 Thank you for valuable suggestion about including to the analysis factors like: type of treatment and type of surgery. W included this variables to analysis in Results part in line 177-179 and in tables 2-5. In discussion we added in line: 247-252 “In our multivariate analysis, next to age >64 years (p=0.04), alone surgery (p=0.02) were found to be associated with an increased risk of death. Generally, neo- or adjuvant therapy could be the most important predictor of better OS, because the most older patients were treated with alone surgery (59%) in compare to younger patients, who received neo- or adjuvant therapy (69%) – this difference was statistically significant (p=0.04)”.

The percentage of extent of surgical resection: R0 vs R1-2 in younger and older group did not significantly differ (p=0.71).

Minor revision:

Ad.1 We changed term “progression-free survival” on term “time to recurrence” and also description of TTR was included in statistical analysis part in line: 105-108.

Ad.2 We agree that the term “average” is not precise used. We changed that on the mean with standard deviation in line 115, 119 in Results and in table 1.

Ad.3 We prepared two tables with hope to be more clear. Table 2 - mOS/mTTR with 95%CI calculated by K-M method and p-value of log-rank test. Table 3 – The percentages of patients with 1-, 3-, 5-year survival from various subgroups compared by Fisher’s exact test. The description of this method is added to statistical analysis part in line: 100-101. The description of results was included in results part in line: 138-166.

Reviewer 4 Report

The given manuscript patient outcomes of LCNEC treated with resection with various stages. The study is well conceptualized with a good distribution of patient data including age, gender, stage, tumor size with this rare form of the lung cancer. The MS can be improvised with the following comments:

1) The study compares the survival using Kaplan Meier survival analysis. The survival analysis should be plotted in the same graph with resection and not resected to understand the comparison clearly. Also, the Y-axis on survival analysis generally represents the % survival. The change can be made in the graph. 

2) In table 1, the last row represents treatment with chemo and without chemo with SCLC and NSCLC is unclear. 

3) In the demographic patient data, the smoking status of patients like heavy/light smokers with pack size is missing, which makes the study less conclusive.

4) The things missing in the MS and can be improvised- Comment about the cellular origin of this type of lung cancer, is it alveolar type I cells? Particular patient inclusion and exclusion criteria are missing, The chemotherapeutic treatment is not significant, maybe because of the type of chemo given to patients? like Platinum-based treatment? Need to be discussed in the discussion. As LCNEC is a locally aggressive form of SCLC, the comment on the comparison with it is appreciated in the discussion.

Author Response

Thank you for a review and valuable comments.

Ad.1 We corrected the figures 1-2 with the % survival on the Y-axis. We provided survival analysis on a group underwent surgery and we did not compare to inoperable group.

Ad.2 In all tables we corrected description of type of chemotherapy: SCLC/NSCLC-chemotherapy to be more clear and in results in line: 120-123. We explained the meaning of  SCLC/NSCLC-chemotherapy in materials and methods part in line: 81-83.

Ad.3 We added smoking status data to characteristic of patients in table 1.

Ad.4. We added in materials and methods part inclusion criteria of patients in line: 88-92.

We added to discussion part the comments of comparison the clinicopathological characteristics and survival outcomes between LCNEC and SCLC in line 259-281. Chemotherapy treatment for LCNEC is a subject of debate since it seems to be less chemosensitive than SCLC. In the American Society of Clinical Oncology (ASCO) guideline, either platinum–etoposide chemotherapy (SCLC-PE) treatment or the same regimen as for non–small cell nonsquamous cell carcinoma is advised for LCNEC, although SCLC-PE is considered as the most appropriate. Nevertheless, recent studies indicate that patients with LCNEC have a more favorable outcome when treated with platinum–gemcitabine or taxane chemotherapy (NSCLC-GEM/TAX) compared with SCLC-PE. We compared survival of patients treated with surgery and chemotherapy regimens derived from SCLC protocols (platinum+etoposide) and NSCLC protocols (platinum+vinorelbine) and  we did not detect a significant difference in treatment results and patient outcome. It could be caused by small size of analyzed subgroups.

Round 2

Reviewer 1 Report

I Have no further comments

Reviewer 4 Report

The authors tried incorporating the modifications suggested by the reviewer. The study is insightful in the field of rare lung malignancy and clinical perspective post-resection.